# Reasons for the Sex Bias in Osteoarthritis Research: A Review of Preclinical Studies

**DOI:** 10.3390/ijms241210386

**Published:** 2023-06-20

**Authors:** Madeline Franke, Chiara Mancino, Francesca Taraballi

**Affiliations:** 1Center for Musculoskeletal Regeneration, Houston Methodist Academic Institute, Houston, TX 77030, USA; mfranke@houstonmethodist.org; 2Orthopedics and Sports Medicine, Houston Methodist Hospital, Houston, TX 77030, USA

**Keywords:** osteoarthritis, sex, gender, gender bias, preclinical studies, men, women, sex as a biological variable, sex difference

## Abstract

Osteoarthritis (OA) is one of the most common degenerative diseases of articular cartilage. During OA, all the elements that contribute to the joint undergo physiological and structural changes that impair the joint function and cause joint pain and stiffness. OA can arise naturally, with the aging population witnessing an increase in diagnoses of this pathology, but the root causes of OA have yet to be identified, and increasing interest is arising towards investigating biological sex as a risk factor. Clinical studies show increased prevalence and worse clinical outcomes for female patients, yet most clinical and preclinical studies have disproportionately focused on male subjects. This review provides a critical overview of preclinical practices in the context of OA, highlighting the underlying need for taking biological sex as both a risk factor and an important component affecting treatment outcome. A unique insight into the possible reasons for female underrepresentation in preclinical studies is offered, including factors such as lack of specific guidelines requiring the analysis of sex as a biological variable (SABV), research-associated costs and animal handling, and wrongful application of the reduction principle. Additionally, a thorough investigation of sex-related variables is provided, stressing how each of them could add valuable information for the understanding of OA pathophysiology, as well as sex-dependent treatment strategies.

## 1. Introduction

Osteoarthritis (OA) is a highly complex joint disease that can cause progressive, permanent structural tissue damage and disability. It affects an estimated 300 million individuals worldwide and its prevalence is increasing due to a rise in important risk factors, such as age and obesity [1,2,3]. The pathophysiology of OA involves mechanisms of perpetual inflammation and chronic joint degradation that can lead to pain, loss of mobility, and irreversible tissue destruction [4,5]. Over time, OA can result in permanent, structural damage to the articular cartilage, surrounding bone, and other components of the synovial joint [1]. Joints that are load bearing or under repeated stress tend to be affected most by OA, namely the knees, hips, and hands [6,7]. Overuse and injury are major causes of OA, with influences from genetics, predisposing risk factors, and anatomical anomalies [8]. Despite being the cause of significant burden in affected individuals, OA has no disease-modifying therapy [1,9]. The mainstay of current treatment revolves around symptom management, pain control, and increasing the quality of life in these patients [10]. As the molecular mechanisms involved in OA development are better understood, more research can be carried out to produce effective therapies focused on tissue regeneration and disease reversal.

An important risk factor in the presentation and progression of OA is biological sex [1,11,12,13,14]. OA disproportionately affects women, as the disease prevalence and experienced pain level is higher in this group than in men [3]. Additionally, the risk of developing OA in women rises substantially after menopause [3]. Prior studies have investigated the reasons for the disparity in OA prevalence and risk between sexes, encompassing biological sex-related factors, genomic variances, transcriptomic variances, and regulatory variances between men and women, but the complexity of the differences in men and women is still not fully understood [3,15]. Tschon et al., 2021 conducted a comprehensive analysis of 42 human osteoarthritis studies focusing on morphometric parameters, kinematics, pain, and functional outcomes post-arthroplasty [3]. Their findings revealed that women exhibit a higher prevalence of osteoarthritis, decreased cartilage volume, increased inflammation and clinical pain, as well as greater healthcare requirements. Another study highlighted a gender bias in the outcomes of a novel therapeutic approach, namely micro-fragmented adipose tissue (MFAT) [16]. Factors such as hormones, anatomical differences, injuries, joint instability, and asymmetric joint loading are just some of the many characteristics that could be contributing to the differences identified in OA between sexes [3,17]. Since OA has a clear propensity to develop in women, there has been a push to better understand the precise mechanisms and intricacies of the disease to establish individualized care for affected patients [17].

Although there has been an increasing interest in the differences between men and women in OA, prior research in the field has focused primarily on male subjects [14,18]. Both clinical and preclinical studies have largely used only male participants, animals, and specimens for their data collection and analyses [3,19]. Though women experience OA differently than men and are at greater risk for the disease, much of the research contributing to therapies and OA treatment is based on the outcomes in men [18]. This is especially a concern in preclinical research, which forms the basis of our understanding of the molecular mechanisms involved in OA pathogenesis and lays the foundation for future work in the field [19] by investigating molecular and histological pathways, progression markers, gene expression, miRNA signatures, and disease-modifying treatments, exploiting established surgically, chemically, or genetically induced disease models. Key problems in preclinical research include the disproportionately high number of male animals used and the lack of any analysis on the experimental differences between the sexes in the studies [18,20]. This underrepresentation of female specimens, especially when researching a condition that affects men and women differently, could alter the understanding of key sex influences on disease pathogenesis and outcomes [20]. Other studies, even with small groups, specify the use of animals from both sexes but do not report sex-specific results [21].

This bias in biomedical research focusing more on male specimens transcends OA research and can be found in many other disciplines, including neuroscience, pharmacology, and immunology [22]. Moreover, the imbalance in males and females in research has been a problem spanning decades. In 1993, the National Institutes of Health (NIH) Revitalization Act called for the inclusion of women in clinical research [23]. Later on, recommendations were proposed in preclinical research to mitigate sex-related bias. For instance, the ARRIVE guidelines (from 2010) emphasize the need to report detailed information on the animals used, including species, strain, sex, and developmental stage, in in vivo experiment results [24]. Years later, in 2014, the NIH further mandated that investigators account for Sex as a Biological Variable (SABV) to better balance the use of male and female specimens in preclinical research [25]. This policy has required all NIH applicants to include SABV in their research design and analysis when studying vertebrate animals or humans [25]. This mandate comes after the informative work of Beery and Zucker which investigated the extent of the sex bias across biomedical research and the implications of the imbalance in various fields [22]. Their work revealed that 8 out of 10 investigated disciplines had a clear male bias and over 60% of reports in certain fields failed to indicate the sexes of the animals used in their experiments [22]. Since then, there has been some improvement, but female underrepresentation in biomedical research is still prevalent in many fields [25].

Currently, there are research gaps in our understanding of sex differences in OA caused in part by the lack of female representation in early preclinical research. Some recent reviews have investigated the gender disparities of prior preclinical work in OA, but they do not offer much insight as to the reason for focusing on male animals or the consequences of doing so. The purpose of this review is to investigate the reasons for the underrepresentation of female specimens in preclinical research and the implications of not analyzing SABV in past studies. We start with an overview of preclinical research across different biomedical disciplines and propose reasons for the focus on male specimens. We then offer a deeper analysis of our topic as it pertains to past and present OA research.

## 2. Sex Bias across the Disciplines

Failing to explore sex differences in biomedical research can be detrimental to patient health yet it happens across many important fields in medicine [22,26]. For example, there are clear biological sex-related differences between men and women in the risk factors, symptom presentation, and prognosis of major cardiovascular diseases, such as myocardial infarctions and strokes [26,27,28]. However, prior cardiovascular research has traditionally included mainly male subjects in both human and experimental models [27,28]. This has contributed to certain setbacks in women’s health, such as the lack of evidence for equally beneficial pharmacological agents and the lack of clear treatment goals for cholesterol and blood pressure for women to follow [26]. Beery and Zucker investigated the sex bias in various topics in biomedical research (Figure 1) and found that a male bias was present in fields ranging from neuroscience and pharmacology to physiology and endocrinology [22]. They looked at six decades of past data to understand the trends over time and determined that the sex was not even specified in a large majority of the early studies [22]. It was not until the late 1960s that the sex of the animal model was made clear in many research articles, and when the shift happened there was a rise in male-only studies [22].

Unfortunately, this bias still exists today. Even when investigating diseases that occur more in women, 44% of the studies did not specify the gender of the specimens and only 12% of the studies used female animals for their analysis [29]. A simple explanation for the reason why women are still underrepresented in biomedical research is that there is a resistance to change within the scientific community [20,23]. Any form of change is difficult, and it might be especially challenging to change conventions when there is no clear need to. In early studies, not understanding the implications of using only male specimens could have prevented researchers from making the effort to change practices in the first place [23]. Additionally, researchers may not have included female specimens because there were initially no guidelines set in place that mandated the representation of both male and females. Before the NIH Revitalization Act of 1993, it was encouraged that women be included in biomedical research, but this was not mandatory. Even after this new Act, which required the appropriate inclusion of women in NIH-funded biomedical research, change was still slow, prompting the NIH SABV policy in 2014 [25]. The NIH is making multiple efforts to address the underrepresentation of women and inadequate consideration of SABV in biomedical research, which has encouraged the development of many other important policies, working groups, and organizations working towards change [25].

Another potential reason for using only male specimens, animals, and cells in preclinical research is the assumption that the data for males could be applied to females. This idea of generalization uses a small number of specimens in one group to then apply the results to a larger group, adopting the belief that the smaller group is an adequate representation of the larger group. This has also allowed researchers to use the minimum number of animals needed for a significant result and then apply the conclusions of the study to an entire population. Researchers may see this as application of the reduction principle, which is considered a humane approach in animal studies because it calls for the use of the fewest amount of animals needed for an adequate study design [20]. However, this is all under the assumption that there are no biological differences between the sexes and that a male-only analysis is sufficient, which is not true [20], and the inclination to minimize the number of research animals to a reasonable minimum may inadvertently introduce gender bias. Another reason researchers might use only male specimens is the idea that adding females increases the expense and complexity of the experiment [20]. In the case of in vivo studies, including more animals means potentially spending more time, precaution, and money on equipment. There are more things to consider when housing the animals, such as if factors such as pheromones or the behavior between males and females may induce unwanted, confounding variation in the results [30]. Certain cases do exist where it is not favorable to include both males and females in the study design. An example of this is a certain murine strain, BALB/c, which has been seen to be associated with strikingly aggressive males leading to utilization of more females [31]. Other than in very specific circumstances, including both sexes in the experimental design and analysis is necessary to apply experimental conclusions on a wide scale.

The pressure to yield statistically significant results may influence how researchers design their experiments. Some of the hesitancy towards including females lies in the idea that females will add unwanted variability to the results [32]. Authors often strive to reduce sexual heterogeneity and hormonal variability in research animals. This trend aligns with the variable-criteria sequential stopping rule (SSR) and others [33], which necessitate a delicate balance between a small, yet statistically justifiable, number of animals in study groups. Nonetheless, this approach poses a notable source of sex-related bias. Additionally, it has been debated whether females are inherently more variable than males because of their hormone cycles, which can negatively impact the reproducibility of data [32]. The argument about the increased inherent variability of females compared to males has been disproven on multiple accounts in various animal studies [34,35,36,37]. Interestingly, significantly greater variance has been seen in males compared to females [36,37]. Additionally, prior research has shown us that female mice are not intrinsically more variable than male mice, and there is no need to study females at different times in their estrous cycle to obtain reliable data [34,35,38]. Therefore, the cyclic estrous cycle experienced by females is not a confounding variable and does not lead to increased variability in the results between sexes. Another issue is that of studies that include an appropriate number of female specimens but fail to analyze SABV. These studies tend to pool the data, grouping all the males and females together instead of analyzing them separately. This is disadvantageous in an analysis because pooling the data can conceal relationships in the data, if any exist [20]. It is not always necessary to find significant differences between males and females if there are none, but analyzing the data separately to check for potential differences is critical for a complete analysis of the data.

Similar problems exist in cellular in vitro studies, due to the fact that the sex of cells is often not specified or considered relevant [39]. In 2013, 75% of the studies published in AJP-Cell Physiology did not specify the sex of the cells in their reports [39]. Moreover, in cases where cell sex was defined, almost 70% used only male cells [40]. Some researchers may even consider cells as asexual entities, but cells do have a defined sex [20,39,41]. Furthermore, cells are often sold from vendors without a defined sex, which adds to the difficulty [42]. It can be challenging as a researcher to make sure that an experiment incorporates both male and female cells if the cell vendors do not take sex into account either. The sex of the cells used in experiments can alter how cells grow, differentiate, and respond to their surroundings [41]. Sex plays as much as a role in the in vitro studies as it does in the in vivo studies. Therefore, careful consideration in selecting cells and reporting their sex in an experimental analysis is of the utmost importance to ensure that females are adequately represented in preclinical in vitro research.

## 3. Sex Differences in Preclinical OA Research

Many of the same reasons for the overall lack of female representation or analysis of SABV across disciplines can be found in OA research. If the goal is to produce research with high reproducibility and statistical significance, researchers may opt not to include females, since there are clear differences in OA initiation and progression between the sexes. In early preclinical studies, the ways in which OA differs in males and females could have been considered as confounding variables that would increase the variance of results. However, it is precisely these differences that are integral to the understanding of the specific mechanisms of OA in each sex (Table 1). Only by taking time to analyze SABV will we recognize how to develop methods of treatment targeted to the pathophysiology of the individual.

### 3.1. The Role of Hormones

The role of hormones in OA is an incredibly important topic that has sparked interest in researchers for decades. Since the early 19th century, there have been accounts linking joint conditions and the start of menopause in women [43,44]. This association between osteoarthritis and hormones, namely estrogen, has become a widely researched topic in recent years. Although the presence of hormones led researchers away from including female specimens in the past, it has proved to be very important in the pathogenesis of OA progression and therefore worth investigating.

Estrogen is the most widely researched sex hormone in the onset and development of OA [43]. There is complexity in the exact role of estrogen in the disease process, since it has been shown to have both protective and aggravating qualities in OA pathogenesis [14,18,19]. Importantly, estrogen helps with cartilage regeneration in females, whereas testosterone has the same effect in males [45]. This demonstrates that there are sex-specific differences in the roles of these hormones in the development of OA [19]. Though estrogen has a regenerative role in women, hormone replacement therapy with exogenous estrogen has not proven to be beneficial in protecting against knee OA [18,46]. Because of the conflicting studies regarding estrogen and the development of OA, more research still needs to be done in this area to fill the gaps in our knowledge [47,48]. Other hormones important to the pathogenesis of OA are progesterone, testosterone, dehydroepiandrosterone, follicle-stimulating hormone, and sex hormone-binding globulin, but their full effects are not entirely clear [49,50,51,52,53].

Testosterone, in particular, has been shown to increase the regenerative capabilities of chondrogenic progenitor cells found in late-stage OA [45]. Considering the physiologically high levels of testosterone in males compared to females, it is understood that testosterone is likely to have a sex-dependent role in cartilage protection and restoration. Interestingly, physiologic levels of testosterone decrease type I collagen gene expression in males but enhance type I collagen expression in females [45]. This is important because type I collagen is not the appropriate collagen subtype for healthy joint cartilage. Thus, testosterone has opposite effects in males and females that eventually promote either healthy or impaired joint tissue. Similarly, a precursor to testosterone, dehydroepiandrosterone (DHEA), also plays a role in various chondroprotective mechanisms [51]. Known effects of DHEA include the suppression of proinflammatory cytokines in the synovial fluid (such as IL-1β and TNF-α), balancing anabolic and catabolic processes, and inhibiting pathways involved in destructive tissue metabolism [51]. Since both of these hormones have the power to interfere with upstream cellular pathways involved in the pathogenesis of OA, they are important to investigate in the context of males and females.

Despite playing critical roles in OA pathogenesis, sex hormones cannot fully be responsible for all of the differences found between male and female OA presentation and progression [14,19]. There are many other structural and molecular differences between the sexes that cannot be attributed to the influence of hormones alone. The importance of adequately representing females in preclinical research regarding the hormones affecting OA is evident, considering that men and women experience different concentrations of hormones at varying times in their lives. Additionally, since the same hormones can have opposing effects in the same type of tissue depending on the sex of the animal, more research can be carried out to investigate the reason for this and the potential benefits of hormone therapy in OA treatment.

### 3.2. Sex Differences in Inflammation and Pain

It has been extensively researched that biologic sex plays a role in inflammation and regulation of the immune system [54]. Although OA is not considered an inflammatory arthropathy compared to rheumatoid or psoriatic arthritis, localized inflammation and chronic cytokine release are part of OA progression. Specifically, joint overuse causes increased biomechanical stress which results in damage to the extracellular matrix at the joint site [19,55]. The degradation of articular cartilage leads to chondrocyte metabolism, igniting a perpetual state of inflammation with the recruitment of cytokines, such as interleukin-1β, tumor necrosis factor-α, IL6, and IL8 [55]. These cytokines stimulate the inflammation and breakdown of tissues, with help form metalloproteinases and prostaglandins [55]. There are many sex-related differences in the inflammatory process of OA. Differences have been noted in the composition of synovial fluid: males had higher levels of metalloproteinases, glycosaminoglycans, and anabolic growth factors, whereas females had higher levels of pro-inflammatory mediators and macrophage stimulators [19]. This is important because it demonstrates how females have increased inflammation in the synovial membrane compared to males when controlling for all other factors. This may contribute to greater debilitation and more severe symptoms in females with OA.

It has been demonstrated that female rats experience a greater level of pain compared to their male counterparts [56]. This, in part, could be due to the elevated inflammatory response they experience and higher level of cytokines in their joint synovial fluid [19]. Not only do females experience more pain, but they also have a greater susceptibility to developing ongoing pain and central sensitization, even with a similar degree of joint pathology to males [57]. These significant differences in inflammation and pain between males and females demonstrate the importance of representing females adequately in biomedical research and analyzing SABV. Similar trends in pain between the sexes have been seen in clinical data, so investigating males and females equally may reveal important mechanisms of pain that could help eventually in guiding treatment or individualized targeted therapy [18].

### 3.3. Patterns of Onset and Progression of OA in Males and Females

There are important sex-related differences in the prevalence and severity of OA in preclinical animal studies. In one study, there was more cartilage loss, reduced bone volume, osteophyte formation, synovial inflammation, and overall progression of OA after a set period in female rats compared to male rats [58]. This demonstrates greater permanent structural changes in the joint that have the potential to lead to a higher degree of functional disability in these animals over time. Moreover, there was reduced gene expression of important growth factors, insulin-like growth factor-1, extracellular and matrix proteins, and IGF binding protein-3 in females [58]. This may indicate that OA progression and severity is increased in females, due to the interplay of genes and the individual inflammatory response of the animal. Similarly, in a baboon animal model, the males were seen to develop early, mild OA symptoms while the females experienced faster, more severe OA progression [59]. Many of these animal studies are limited, as some may have a very small sample size or the results have not been reproduced in the same species. Additionally, there are other animal models that provide conflicting results, so additional research is needed in this area to draw more reliable conclusions. However, the preliminary results are encouraging, as they demonstrate potential differences in the structural change and severity of OA between the sexes. These structural differences identified in preclinical studies echo what we know about anatomical differences seen between human males and females in the context of OA development. For example, women tend to have thinner articular tissues, an inherent imbalance in mechanical loading, and a propensity for varus malalignment [17]. These changes over time can lead to uneven weight bearing on a joint which can be a catalyst for the eventual development of OA. Small biomechanical differences between males and females can irreversibly alter a joint and cause characteristic structural differences seen in preclinical studies.

With this information, therapeutic solutions could be designed to target the specific molecular mechanisms that are found in only males or females. Device development could focus on counteracting the structural alterations that occur in either a male or female joint. More work could be carried out to investigate and intervene with an individual’s pathologic gene expression. Working to uncover more of these underlying mechanisms contributing to sex-differences in OA could alter how we approach treating the disease on an individual level.

### 3.4. Implications of Sex Bias and Current State of SABV in Preclinical Research

Today, there are still research gaps in the sex-related mechanisms of OA, in part due to the lack of reproducible preclinical studies that analyzed females or SABV [60]. Consequences of underrepresenting females in preclinical research, pooling data, and failing to analyze SABV in OA research include the limited knowledge we have regarding sex-differences in hormone-influence, arthritic pain, and intrinsic molecular mechanisms responsible for significant disparities in the prevalence and progression of OA in the sexes. On a broader scale, the sex bias in early preclinical research across disciplines has led to a maintained bias decades later, influencing our understanding of how various diseases present and develop in women. Additionally, it has slowed the progression of female-specific therapies and interventions, since much of the important pharmacologic data is tested in only males [22]. While there has been considerable progress in the inclusion and analysis of women, led by the efforts of the NIH, there is still work to be done [20].

## 4. Conclusions

In summary, the underrepresentation and lack of analysis of female specimens in preclinical biomedical research is a problem that has existed for a long time. It especially impacts fields such as osteoarthritis where there are clear, important differences between male and females that need to be investigated. With the help of the NIH mandates, the adequate representation of females and the analysis of SABV is an issue that has been brought to the attention of the scientific community. While current gaps in our knowledge of sex-differences in OA prevalence and pathogenesis still exist, widespread collaboration in an effort to change has the power to alter the state of future research for the better.

## Figures and Tables

**Figure 1 ijms-24-10386-f001:**
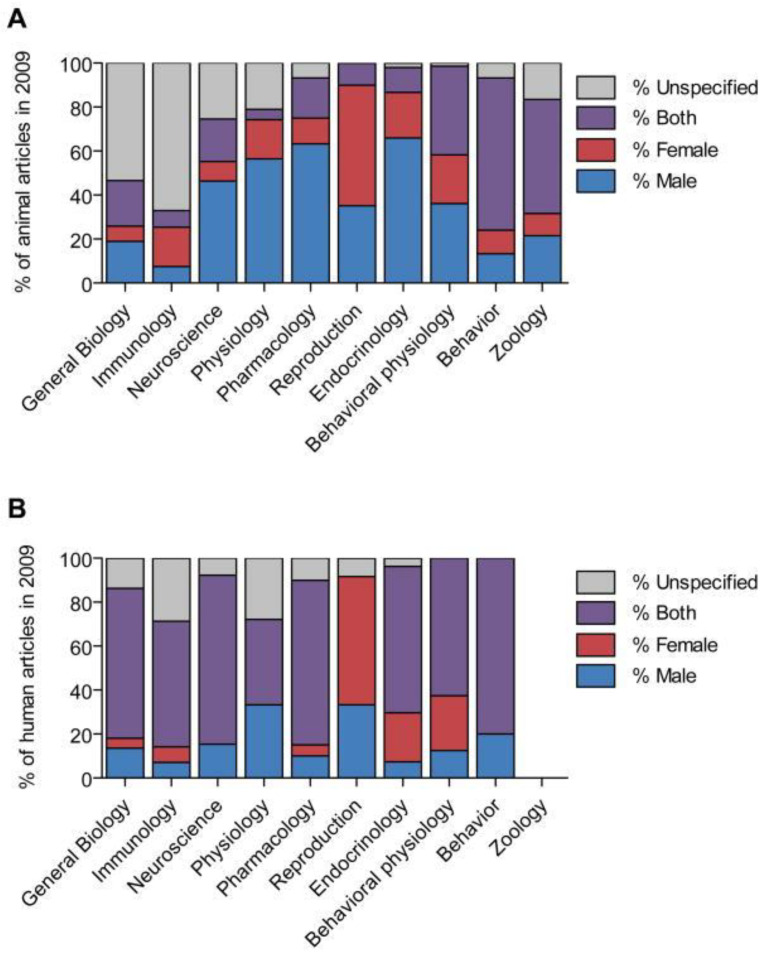
Distribution of studies by sex and field in 2009. (**A**) Percentage of articles describing non-human animal research that used male subjects, female subjects, both male and female subjects, or did not specify the sex of the subjects. (**B**) Percentage of articles describing human research in the same categories. Adapted from Beary et al., 2011 [22].

**Table 1 ijms-24-10386-t001:** Summary of preclinical studies in osteoarthritis research that investigate differences in sex.

Area of Focus	Objective	Main Conclusions	References
Differences in hormones	Role of estrogen	Estrogen is both protective and aggravating in OA pathogenesis. In females, estrogen helps to regenerate cartilage, thereby interfering with disease progression. The same restorative qualities of estrogen have not been seen in males.	[14,18,19,43,44,45,46,47]
	Role of testosterone	Testosterone has been seen to have restorative qualities in males. In females, higher testosterone levels are associated with lower joint pain scores. Lower testosterone leads to increased pain sensitivity.	[48,49]
	Role of progesterone	Progesterone has been seen to have anti-inflammatory effects in OA due to its gene suppression of cytokine production.	[50]
	Role of dehydroepiandrosterone	Dehydroepiandrosterone (DHEA) has both protective and aggravating functions in OA progression. The overall effect of the hormone is not clear. It helps to modulate the balance between cartilage anabolism and catabolism. It can decrease remodeling of the subchondral bone to slow joint space deterioration.	[51]
	Role of follicle-stimulating hormone	Follicle-stimulating hormone, a hormone higher in women, works to promote the inflammatory response of chondrocytes. Introduction of FSH into the joint space can worsen OA.	[52]
	Role of sex hormone-binding globulin	Increased levels of sex hormone-binding globulin (SHBG) positively contribute to the development of OA. Additionally, high levels of this hormone could be considered a risk factor for OA. The exact mechanism of SHBG in OA is still being investigated.	[53]
Differences in pain and inflammation	Pain response	Females tend to experience greater levels of pain than males and are more susceptible to developing chronic pain.	[54,55]
	Inflammation and the immune system	Females have a heightened inflammatory response with higher levels of pro-inflammatory cytokines and macrophage stimulators. Males have increased anabolic growth factors, metalloproteinases, and glycosaminoglycans in their synovial fluid and less inflammatory mediators. In the same joint, females and males can experience different levels of inflammation which drives pain and debilitation.	[56,57]
Differences in disease characteristics	Joint anatomical changes, biomechanical factors, disease progression	When controlling for time, females experience more cartilage loss, reduced bone volume, increased osteocyte formation, and a higher degree of OA progression and severity than males. This leads to greater functional disability in females in a shorter amount of time than males.	[58,59]
	Gene expression, presence of growth factors	Females have reduced gene expression of growth factors (insulin-like growth factor-1, extracellular matrix proteins, IGF binding protein-3) compared to males.	[58]

## Data Availability

Not applicable.

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
