# Peer review of "Reasons for the Sex Bias in Osteoarthritis Research: A Review of Preclinical Studies"

_ijms, 2023, doi:10.3390/ijms241210386_

Round 1
Reviewer 1 Report
This is a well-written, interesting review from Franke M. et al. addressing the causes, motivations, and effects of sex bias in osteoarthritis research. The scientific community learned in the past decade that there are many concrete aspects concerning the biological sex-related susceptibility, genomic, transcriptomic, and regulatory differences in osteoarthritis of men and women. For example, Tschon M. et al. analyzed 42 human osteoarthritis studies for morphometric parameters, kinematics, pain, and functional outcomes after arthroplasty. They found that women have higher OA prevalence, decreased cartilage volume, more inflammation and clinical pain, and higher healthcare needs [1]. Another study revealed that a gender bias exists in the outcomes of a new therapeutic procedure, namely the micro-fragmented adipose tissue (MFAT) treatment [2].
Many aspects of this complex disease are investigated at the preclinical level: molecular and histological pathways, progression markers, gene expression, miRNA signature, and disease-modifying treatments. Moreover, several different surgically, chemically, or genetically induced disease models exist. Since the 2014 NIH principles were developed, these can be used as guidelines to use sex as a biological variable (SABV) in scientific research.
In the preclinical research, there were some recommendations to avoid sex-related bias even before the elaboration of SABV. For example, the ARRIVE guidelines state that in reporting results of in vivo experiments, researchers should “provide details of the animals used, including species, strain, sex, developmental stage”[3]. Despite this, as Franke et al. mention, “clinical and preclinical studies have largely used only male participants, animals, and specimens for their data collection and analyses” and extrapolate or negligent female osteoarthritis. Other studies, even for small groups, specify the application of animals from both sexes but do not report sex-specific results [4].
On the other, it has to be understood that the tendency to reduce the number of research animals to a reasonable minimum may generate a gender bias since many authors try to avoid sexual heterogeneity and hormonal variabilities in research animals. The variable-criteria sequential stopping rule (SSR) [5] and other recommendations act in this direction, and authors have to balance between a small, homogenous, but at the same time, statistically reasonable number of animals in their study groups. This is an important source of sex-related bias.
The quality of this manuscript would be improved with the embedding of these observations. Also, I recommend to complete the list of references with those indicated below.
With these modifications, I agree with the publication of the manuscript elaborated by Franke M. et al. in International Journal of Molecular Sciences.
References
1. Tschon, M.; Contartese, D.; Pagani, S.; Borsari, V.; Fini, M. Gender and Sex Are Key Determinants in Osteoarthritis Not Only Confounding Variables. A Systematic Review of Clinical Data. J. Clin. Med. 2021, 10, 3178. https://doi.org/10.3390/jcm10143178
2. Borg TM, Heidari N, Noorani A, Slevin M, Cullen A, Olgiati S, Zerbi A, Danovi A, Wilson A. Gender-Specific Response in Pain and Function to Biologic Treatment of Knee Osteoarthritis: A Gender-Bias-Mitigated, Observational, Intention-to-Treat Study at Two Years. Stem Cells Int. 2021 Feb 25;2021:6648437. doi: 10.1155/2021/6648437
3. Kilkenny C, Browne WJ, Cuthill IC, Emerson M, Altman DG (2010) Improving Bioscience Research Reporting: The ARRIVE Guidelines for Reporting Animal Research. PLoS Biol 8(6): e100041
4. Csifo EN, Nagy EE, Horvath EM, Farr A, Muntean DL. Mid-term effects of meloxicam on collagen type II degradation in a rat osteoarthritis model induced by iodoacetate. Farmacia. 2015;63(4):556-60.
5. Fitts DA. Minimizing animal numbers: the variable-criteria sequential stopping rule. Comp Med. 2011 Jun;61(3):206-18
Author Response
The authors thank the reviewer for the valuable observations made, and have edited the manuscript introducing all the necessary modifications, as well as the suggested citations. Observations have been embedded in the introduction (lines 46-56, lines 67-72, lines 77-79, lines 84-88, lines), and in paragraph 2 (lines 156-160, lines 173-177).
Reviewer 2 Report
1. I suggest added a brief discussion about testosterone, and dehydroepiandrosterone in the pathogenesis of OA.
2. Anatomical & Biomechanical factors needs be mentioned.
3. Please add a table to summarized section 3: sex differences in preclinical OA research.
Author Response
The authors thank the reviewer for the valuable suggestions, and have edited the manuscript accordingly, by introducing a table summarizing sex-related differences in OA (page 6), as well as a brief discussion about testosterone, and dehydroepiandrosterone in the pathogenesis of OA (lines 244-258), and a comment on anatomical and biomechanics factors (lines 314-321).